# Factors that influence safe water drinking practices among older persons in slums of Kampala: Analyzing disparities in boiling water

**Fred Maniragaba** [1] *, **Abel Nzabona** [2], **Charles Lwanga** [1], **Paulino Ariho** [1], **Betty Kwagala** [1]

**1** Department of Population Studies, Makerere University, Kampala, Uganda, **2** Center for Basic Research, Kampala, Uganda

* fmaniragaba@gmail.com

**Editor:** D. Daniel, Gadjah Mada University Faculty of Medicine, Public Health, and Nursing: Universitas Gadjah Mada Fakultas Kedokteran Kesehatan Masyarakat dan Keperawatan, INDONESIA

## Abstract

### Introduction

Safe drinking water practice is a key public health promotion strategy for reducing the spread of waterborne diseases. The purpose of this study was to investigate the factors that influence boiling water practice among older persons in informal settlements of Kampala.

### Methods

We collected primary data on *"Access to safe water and health services among older persons* in informal settlements of Kampala in October 2022. The study interviewed 593 men and women aged 60 years and older. The Complementary log-log multivariable technique was used to establish the association between boiling water practice and selected independent variables.

### Results

The results show that nearly 8% of the respondents did not boil their water for drinking. The findings show that the female older persons had increased odds of boiling water to make it safe for drinking compared to their male counterparts (OR = 1.859, 95% CI = 1.384–2.495). Other factors associated with boiling water practice among older persons in the informal settlements of Kampala were; living alone, quality of house, and type of water source.

### Conclusion

Basing on our findings, we find that older women are more likely to use safe drinking water practice (boiling) compared to the male older persons. Health education majorly targeting older men about the importance and health benefits associated with safe water drinking practices should be prioritized by policy makers. There is need to improve housing conditions of older persons to minimize typhoid, diarrhea and other health risks associated with drinking unsafely managed water.

**Data Availability Statement:** All relevant data are within the paper and Supporting information files.

**Funding:** "The authors received no specific funding for this work".

**Competing interests:** The authors have declared that no competing interests exist.

## Introduction

Access to safe drinking water and sanitation is an important basic need that stimulates good health among the people and also, enables them fully enjoy their life [1]. It is a human right for both young and older persons [2,3]. This right demonstrates an increasing global commitment to ensuring universal access to water for domestic use enshrined in both the Millennium Development Goals (MDGs) and the Sustainable Development Goals (SDGs) [4]. The 6th Sustainable Development Goal aims at ensuring availability and sustainable management of water for all by 2030 [5,6]. Safe drinking water is therefore, required to sustain life and human health [7,8]. In the year 2020, 74% of the world's population (5.8 billion people), which was an increase from 62% (3.8 billion) in 2000 [9], used safely managed drinking water.

In spite of the progress made on the access to safe water, the world is not close to being on track to meeting the 2030 SDG goals [1]. Globally, an estimated 2 billion people lack access to safe drinking water. Moreover, 2 in every 5 persons in urban areas, do not have safe water for drinking [10]. In addition, billions of people, both old and young, in low and middle-income countries (LIMCs), continue to lack access to adequate safe water, sanitation, and hygiene (WASH) services [11]. This situation could be worse in informal settlements of urban areas in sub Saharan Africa, where most vulnerable people, including older persons live. Thus, access to safe drinking water and improved sanitation remains one of the major issues confronting developing countries [12]. In low and middle income countries, some people rely on water sources contaminated with fecal bacteria for drinking water, and there is hardly any boiling practice done to make it safe [13,14]. Safe drinking water is defined as water that does not represent any significant risk to health over a lifetime of consumption [16]. In sub-Saharan Africa (SSA), boiling is the common practice used in making water safe for drinking [14].

World Health Organization (WHO) and United Nations Children Emergency Fund (UNICEF) reports have documented fecal contamination of drinking water sources, especially in the slums of the urban area where some people draw water and drink it without first boiling to make it safe [9]. Consequently, water-related diseases among the vulnerable people especially the older persons, remain the primary cause of the high mortality rate in developing countries [15,16]. For example, unsafe drinking-water accounts for 485,000 diarrhea deaths each year [13,14,17], in low and middle-income countries. In the same vein, with a rapid population growth in sub-Saharan Africa, the problem of access to safe water for drinking is likely to worsen.

In the year 2020, 411 million people in Africa, lacked basic drinking water, and 3 out of 5 people living in Africa's urban areas lacked safely managed drinking water [10]. Relatedly, projections indicate that only 37% of the population in sub-Saharan Africa will be using safely managed drinking water by 2030 [9]. This problem will be worse in the urban areas as the sub region continues to experience rapid urbanization dominated by growth of slums, which are also usually the home for the many vulnerable people such as older persons [18]. Notwithstanding the challenges such as poor access to safe water and hygiene faced by older persons and other slum dwellers [19–22], preventing the spread of waterborne diseases like, typhoid, dysentery, cholera and diarrhea through drinking water boiling practices is important to their well-being [9,23]. Studies show that these water borne diseases put the health and lives of many people at a great risk [24].

Thus far, before use, much of the water for drinking requires treatment options such as boiling, filtration, chlorination, solar disinfection and covering of water storage containers to reduce the risk of contamination from dust or other airborne particles [14,25–27]. These treatment options improve the quality of drinking water and also effectively reduce morbidity and mortality among the users [28]. Whereas the treatment practices improve the safety of

drinking water, storage facilities may re-contaminate the treated water at the point of use [29]. Re-contamination may also arise from the nature of utensils used for drinking water such as cups, bowls, and other containers [13,30,31]. However, the challenge with some of the above water treatment options is that, they are relatively costly for most households.

Boiling is the relatively affordable household water treatment method for water quality improvement and when consistently and effectively used, it drastically reduces fecal water contamination [24]. This water treatment technology has however been given little attention by researchers. There is dearth of boiling water-based research especially in areas that are prone to contamination, relative to other methods such as filtration and chlorination. Due to social, economic and physical vulnerabilities, older persons defined as persons aged 60 years and above [3], in informal settlements, may face heightened risk of water-related diseases and deaths. Older persons that live in unsanitary environments such as informal settlements are a vulnerable people who are susceptible to waterborne diseases [32]. Even when the first target under SDG 6, is to achieve universal and equitable access to safe and affordable drinking water for all by 2030 [33], older person's access to safe drinking water may not be guaranteed.

A number of studies on drinking water in Uganda have focussed on; adherence to safe water chain practices among the refugees [34]; and water quality of improved water sources [35]. Others have focussed on the factors for viable rural water services [36]; and contamination potentials water handling [37]. In contrast, less research effort has been made to better understand the quality of life of older persons through the lens of health-related practices. Despite their resourcefulness and unique experiences in nation building [38]; severe later-life health challenges still abound among older persons [39–41]. However, research on health-related areas, such as safe water drinking practices among older persons, is limited in Uganda's urban informal settlements. This study therefore, investigated the factors that influence older persons' safe water drinking practices in informal settlements of Kampala.

## Materials and methods

We utilized data from a cross sectional study on the determinants of access to safe water and health services by older men and women aged 60 and older from the informal settlements of Kampala, Uganda. The study targeted 600 respondents and successfully interviewed 593 of them. This translates into a response rate of 98.8%. Eligibility for participation in the study required a participant to be a male or female older person and residing in Kampala's informal settlements (slums). Data collection was done using a structured questionnaire designed from existing survey data collection tools [42,43]. The questions were based on the core questions on safe drinking water; and sanitation and hygiene for household surveys [42]. The questionnaire was programmed on electronic tablets for computer assisted data collection. The data were collected in October 2022. Enumerators with minimum qualification of a Bachelor's degree were recruited from the study areas within the various divisions of Kampala. These were trained on how to handle data collection exercise, and also, on how to observe research ethics while conducting interviews. During the process of data collection, older persons who were sick were excluded from the study. Selection of the study sample involved the use of multi-stage cluster random sampling. At the first stage, three divisions (Kawempe, Nakawa and Rubaga) were randomly selected from Kampala city. At the second stage, 4 parishes were randomly selected out of 19 parishes in Kawempe division, 3 were selected out of 13 in Rubaga division and 5 were selected out of 23 in Nakawa division. At the third stage, zones were randomly selected from each parish. Lastly, using community leaders of the older persons, households were randomly selected from the selected zones. Selection of the households was based

on a sampling frame provided by a community leader. The older persons who consented to participate in the study were interviewed.

## Variables

The outcome variable for this study was safe water drinking practice which was measured based on the question of whether older persons did anything to make their drinking water safe with a probe on the practice of boiling water for drinking. The responses were coded as "boil" for those who reported that they boiled water to make it safe for drinking and "does not" for otherwise.

The explanatory demographic variables were sex categorized as male or female; and age categorized as 60–69, 70–79 and 80+ years. Socio-economic explanatory variables were Education level (no education, primary and secondary+), Occupation (farming, business/trade, dependent, casual work and other); Religion (Catholic, Anglican, Pentecostal and Muslim), Division of residence (Kawempe, Rubaga and Nakawa) and Residential status in household (usual member, regular member and guest). A usual member is a person who has been living in the household for the last six month or more during the last 12 months before our survey. A 'regular' member is a person who would have been a usual member, but has only stayed for less than six months in the household at the time of our survey. Other socio-economic variables were Relationship with household head (head, spouse, children and others); Lives alone in the household (yes or no); Rent status of respondent (renting or not renting); Quality of the dwelling unit (well-maintained, fairly maintained and dilapidated); Household's main source of information (radio, word of mouth, telephone and television); Financial support (gets financial support or does not get financial support); Membership of Association (belonging to Association or not belonging to Association); and Type of water source (improved source or unimproved source).

## Statistical analysis

We used STATA software version 15 to do statistical analyses. Frequency distributions were computed to describe background characteristics of the respondents. At bivariate level, Pearson Chi-squared test, set at $p < 0.05$, was used to establish the association between the dependent variable (measured by whether or not the respondent was doing something to make their drinking water safe) and each of the independent variables. The Complementary log-log multivariable technique was used for this analysis because of the high proportion of older persons that reported that they boiled water to make it safe for drinking. The data were weighted prior to analysis to account for under and oversampling of older persons in the 3 divisions. Since study samples in the three divisions of Kampala, were not uniform due to the size of the divisions, data weighting was done to correct imperfections that would result from the selection of the above divisions with unequal probabilities, mainly to minimize bias in the study findings. A weighting variable was generated basing on the probabilities of selecting; a division from Kampala, parish from a selected division, a zone from a selected parish; and a household from a selected zone. The fitted model was subjected to the *link-test* to examine whether the explanatory variables were appropriately specified. The test uses the *hat* and *_hat-squared* statistic. When the model describes the data correctly and is appropriate, the *hat-squared* is not expected to be significant (*_hat-squared*, *p>0.05*), which implies that the observed data mirror the expected. Before fitting the model, a multi-collinearity test among independent variables (results not presented) was conducted. The variables that were significant at bivariate level were analyzed at the multivariate level. Nonetheless, those that were not significant at bivariate such as age, but deemed to be relevant to socio-economic status of older persons as informed by prior studies were included in the model.

# Results

## Background characteristics of the respondents

Table 1 presents the distribution of respondents by selected socio-demographic factors. The results show that most (92.3%) of the respondents reported that they boiled the water to make it safe for drinking. Nearly two-thirds (65.0%) of the respondents were aged 60–69 years, 71.4% were females. Less than a half (48.1%) of the older persons had completed primary education. Slightly more than five in every ten (50.8%) reported that their occupation was business. In terms of religious affiliation, nearly one-third (32.7%) of the respondents were Catholics, 29.6% were Anglicans while 23.9% were Muslims. A large proportion (34.8%) of the respondents were from Kawempe divisions, Nakawa (33.5%) and Rubaga (31.7%) respectively. More than three quarters (77.4%) of the older persons were usual members of households, 79.4% were heads of households, 88.5% were not living alone, and 73.9% were not renting the houses in which they were staying, while 74.0% were staying in well-maintained houses. Regarding the self-reported source of information, a large percent (35.0%), reported that their main source was television. More than half (52.5%) of the respondents were getting financial support while nearly three quarters (74.1%) were not members of any groups or associations. The table further shows that most (95.1%) of the respondents got their water from an improved source.

## Differentials in safe water drinking practices among older persons by sociodemographic characteristics

Table 2 presents bivariate results for the association between safe water drinking practices and selected socio-demographic factors. Age of the respondents, education level, religion, occupation, division of residence, residential status, relationship with the household head, house renting, household's main source of information, gets financial support and membership of association were not significantly associated with safe water drinking practice (water boiling). The results in the table show that the prevalence of not using safe drinking water practice (boiling water for drinking) among the older persons was higher among the older men (16.1%; $p<0.001$), those who were living alone (17.4%; $p = 0.007$), those living in dilapidated houses (25.4%; $p = 0.001$); and those who were using unimproved source of water (24.7%; $p = 0.004$).

## Predictors of safe drinking water practices by older persons

The findings in Table 3 indicate that the practice of boiling water to make it safe for drinking was significantly associated with sex of older persons, living alone, quality of house, and type of water source. The findings specifically reveal that female older persons had increased odds of boiling water to make it safe for drinking compared to their male counterparts (OR = 1.859, 95% CI = 1.384–2.495), those who were affiliated to Catholics religion were nearly two times more likely to boil water for drinking compared to their Muslim counterparts (OR = 1.712, 95% CI = 1.134–2.587). The table further shows that the odds of boiling drinking water were high among those who were not living alone compared to those who were living alone (OR = 1.759, 95% CI = 1.200–2.578). On the other hand, Older persons who were living in dilapidated houses were less likely to boil water for drinking relative to those who were living in well and fairly maintained houses (OR = 0.455, 95% CI = 0.286–0.725). The findings also reveal that older persons who were getting their water from unimproved sources were less likely to boil it for drinking compared to those who were getting it from improved sources (OR = 0.584, 95% CI = 0.346–0.985).

**Table 1. Distribution of respondents by selected characteristics.**

| Characteristic | Frequency (n = 593) | Percent (%) |
|---|---|---|
| **Boiling water for drinking** | | |
| Doesn't boil | 46 | 7.7 |
| Boil | 547 | 92.3 |
| **Age** | | |
| 60–69 years | 385 | 65.0 |
| 70–79 years | 128 | 21.6 |
| 80+ years | 80 | 13.4 |
| **Sex** | | |
| Male | 170 | 28.6 |
| Female | 423 | 71.4 |
| **Education level** | | |
| None | 113 | 19.1 |
| Primary | 285 | 48.1 |
| Secondary+ | 195 | 32.8 |
| **Occupation** | | |
| Farming | 51 | 8.6 |
| Business/trade | 301 | 50.8 |
| Dependent | 189 | 31.9 |
| Casual work | 31 | 5.2 |
| Other | 21 | 3.6 |
| **Religion** | | |
| Catholic | 194 | 32.7 |
| Anglican | 176 | 29.6 |
| Pentecostal | 82 | 13.8 |
| Muslim | 142 | 23.9 |
| **Division of residence** | | |
| Kawempe | 206 | 34.8 |
| Rubaga | 188 | 31.7 |
| Nakawa | 199 | 33.5 |
| **Residential status in household** | | |
| Usual member | 459 | 77.4 |
| Regular member | 117 | 19.8 |
| Guest | 16 | 2.8 |
| **Relationship with household head** | | |
| Head | 471 | 79.4 |
| Spouse | 60 | 10.2 |
| Children | 44 | 7.4 |
| Other | 18 | 3.0 |
| **Lives alone in the household** | | |
| Lives alone | 68 | 11.5 |
| Lives with others | 525 | 88.5 |
| **Rent status** | | |
| Renting | 155 | 26.1 |
| Not renting | 438 | 73.9 |
| **Quality of the house** | | |
| Well-maintained | 439 | 74.0 |
| Fairly maintained | 116 | 19.5 |

(*Continued*)

**Table 1.** (Continued)

| Characteristic | Frequency (n = 593) | Percent (%) |
|---|---|---|
| Dilapidated | 38 | 6.4 |
| **Household's main source of information** | | |
| Radio | 201 | 33.9 |
| Word of mouth | 75 | 12.7 |
| Telephone | 109 | 18.3 |
| Television | 208 | 35.0 |
| **Financial support** | | |
| Gets financial support | 311 | 52.5 |
| Does not get financial support | 282 | 47.5 |
| **Membership of association** | | |
| Member | 154 | 25.9 |
| Not member | 439 | 74.1 |
| **Type of water source** | | |
| Improved source | 564 | 95.1 |
| Unimproved source | 29 | 4.9 |

## Discussion of the findings

The main objective of the study was to examine safe water drinking practices among older persons in slums of Kampala. The findings indicate that 8% of the older persons in informal settlements did not boil water for drinking. The practice of boiling water for drinking among the older persons was positively associated with; being female, catholic, not living alone. However, living in dilapidated house, and getting water from unimproved water source was negatively associated with boiling water practice.

Our study findings show that female older persons had twice the odds to practice procedures of improving drinking water compared to their male counterparts. This finding is in line with studies done in China [44] and sub-Saharan Africa [45], which revealed that in most households, females were more likely to be the ones that boiled water for drinking compared to males. In the context of the African setting, this could be attributed to the fact that women are known to take on responsibilities of managing households by undertaking various chores such as boiling and storing water safely with minimal contamination risks than males [46]. This finding is also in conformity with literature from studies done in Zambia [47] and Ghana [48], which show that women pay more attention to household issues, including managing water safely for drinking, than their male counterparts. Basing on this substantial evidence on gender disparities in safe water drinking practices, intervention targeting older men's involvement in safely managing drinking water should be underscored to minimize associated health risk and mortality resulting from drinking unsafe water.

We also find that older persons who said that they were Catholics were more likely to boil their water for drinking as a practice to make it safe. This may point to the role that religious institutions play in health education of their followers. For example, the Catholic Church usually conducts health education and sensitization of their congregants during mass, and also, visit them regularly when they are ill [49]. Further to the above, similar studies indicate associations between religion and health [50–53]. Although these studies have not directly examined differentials in safe water drinking practices, their analyses shed a little bit of light on the relevance of religion in public health [54]. In addition, studies such as that done by Olivier, Tsimpo [53], posit that there is diverse evidence to support the idea that faith-based health

**Table 2. Safe water practice by older persons' socio-demographic characteristics.**

| Characteristic | Frequency (n = 593) | Boiling water for drinking | | |
|---|---|---|---|---|
| | | Doesn't boil (%) | Boil (%) | χ2 (p-value) |
| **Age (years)** | | | | |
| 60–69 | 385 | 7.8 | 92.2 | 0.166 (0.938) |
| 70–79 | 128 | 7.0 | 93.0 | |
| 80+ | 80 | 8.5 | 91.5 | |
| **Sex** | | | | |
| Male | 170 | 16.1 | 83.9 | 23.515(<0.001) |
| Female | 423 | 4.3 | 95.7 | |
| **Education level** | | | | |
| None | 113 | 8.5 | 91.5 | 7.411(0.060) |
| Primary | 285 | 4.8 | 95.2 | |
| Secondary+ | 195 | 11.5 | 88.5 | |
| **Religion** | | | | |
| Catholic | 194 | 5.5 | 94.5 | 2.433(0.600) |
| Anglican | 176 | 9.0 | 91.0 | |
| Pentecostal | 82 | 10.2 | 89.8 | |
| Muslim | 142 | 7.6 | 92.4 | |
| **Occupation** | | | | |
| Farming | 51 | 1.5 | 98.5 | 11.149(0.083) |
| Business/trade | 301 | 6.0 | 94.0 | |
| Dependent | 189 | 9.7 | 90.3 | |
| Casual work | 31 | 18.6 | 81.4 | |
| Other | 21 | 13.4 | 86.6 | |
| **Division of residence** | | | | |
| Kawempe | 206 | 6.4 | 93.6 | 1.410(0.536) |
| Rubaga | 188 | 7.2 | 92.8 | |
| Nakawa | 199 | 9.5 | 90.5 | |
| **Residential status** | | | | |
| Usual member | 459 | 7.4 | 92.6 | 5.112(0.161) |
| Regular member | 117 | 6.9 | 93.1 | |
| Guest | 16 | 22.3 | 77.7 | |
| **Relationship with household head** | | | | |
| Head | 471 | 8.2 | 91.9 | 4.244(0.377) |
| Spouse | 60 | 2.6 | 97.4 | |
| Children | 44 | 6.5 | 93.5 | |
| Other | 18 | 16.3 | 83.8 | |
| **Living alone** | | | | |
| Lives alone | 68 | 17.4 | 82.6 | 10.147(0.007) |
| Lives with others | 525 | 6.4 | 93.6 | |
| **Rent status** | | | | |
| Renting | 155 | 11.7 | 88.3 | 4.621(0.060) |
| Not renting | 438 | 6.3 | 93.7 | |
| **Quality of house** | | | | |
| Well-maintained | 439 | 6.4 | 93.6 | 17.994(0.001) |
| Fairly maintained | 116 | 6.9 | 93.1 | |
| Dilapidated | 38 | 25.4 | 74.6 | |
| **Household's main source of information** | | | | |

*(Continued)*

**Table 2.** (Continued)

| Characteristic | Frequency (n = 593) | Boiling water for drinking | | |
|---|---|---|---|---|
| | | Doesn't boil (%) | Boil (%) | χ2 (p-value) |
| Radio | 201 | 9.5 | 90.5 | 9.230(0.053) |
| Word of mouth | 75 | 14.4 | 85.6 | |
| Telephone | 109 | 4.4 | 95.6 | |
| Television | 208 | 5.2 | 94.8 | |
| **Gets financial support** | | | | |
| Gets financial support | 311 | 8.3 | 91.7 | 0.287(0.646) |
| Does not get financial support | 282 | 7.1 | 92.9 | |
| **Membership of association** | | | | |
| Member | 154 | 4.4 | 95.6 | 3.187(0.107) |
| Not a member | 439 | 8.9 | 91.1 | |
| **Type of water source** | | | | |
| Improved source | 564 | 6.8 | 93.2 | 12.489(0.004) |
| Unimproved source | 29 | 24.7 | 75.3 | |

providers continue to play a part in health provision, especially in fragile health systems. Other studies indicate that churches in Africa make a critical practical health contribution in addition to the spiritual role, and argue that churches could operate in communities within the interface of church and health spaces [52,55,56]. Relatedly, Allegranzi, Memish [50], argue that religion has significantly contributed to healthcare ethics by putting emphasis on people's physical and spiritual aspects. Notwithstanding available insights on linkages between religion and health, the bigger picture between the two remains grey and hazy. We therefore recommend that further studies be done particularly on the association between religion and safe water drinking practices by older persons.

Living with others in the household was positively associated with boiling water for drinking among the older persons. Several factors could have combined to positively influence the practice of boiling water among older persons who lived with other people. It could have been the case, for example, that there were shared roles among household members, with some taking on the responsibility, alongside other tasks, of boiling household drinking water. For instance, studies done in Pakistan [57], and other countries [45,58], show that the size of the household determines the quality of drinking water used by its members. Specifically, the small household size is associated with improved drinking water [45]. Further to the above, it is also probable that there may have been certain household members whose health standards were high and thus who required that all drinking water be boiled before consumption. Whereas some persons who live alone, especially the working class, may have struggled to find time for doing household chores, the situation may have been different for their counterparts with different living arrangements. The latter could have had relatives who stayed at home for much of the day and therefore better able to find time for improving the safety of drinking water. Studies have underscored linkages between living arrangements and wellbeing of older persons. For example, older persons who lived in multigenerational households were found to have the lowest levels of short-term illness [48,59]. Among them, those who lived with their spouse, adult children, and young grandchildren experienced the highest health gains. The health advantage diminished when older adults lived only with a spouse and adult children, and further diminished when they lived only with their spouse. In stark contrast, living alone was associated with the highest likelihood of short-term morbidity. Other studies have

**Table 3. Predictors of safe water drinking practices.**

| Characteristic | Odds Ratio | P-value | 95% CI |
|---|---|---|---|
| **Age (years)** | | | |
| 60–69 | 1.000 | | |
| 70–79 | 1.003 | 0.986 | 0.712–1.414 |
| 80+ | 0.903 | 0.616 | 0.606–1.346 |
| **Sex** | | | |
| Male | 1.000 | | |
| Female | 1.859 | *<0.001* | 1.384–2.495 |
| **Education level** | | | |
| None | 1.000 | | |
| Primary | 1.305 | 0.103 | 0.948–1.797 |
| Secondary+ | 0.876 | 0.472 | 0.611–1.256 |
| **Occupation** | | | |
| Farming | 1.000 | | |
| Business/trade | 0.853 | 0.618 | 0.457–1.593 |
| Dependent | 0.658 | 0.200 | 0.347–1.248 |
| Casual work | 0.686 | 0.367 | 0.302–1.557 |
| Other | 0.596 | 0.277 | 0.234–1.517 |
| **Religion** | | | |
| Muslim | 1.000 | | |
| Catholic | 1.712 | *0.011* | 1.134–2.587 |
| Anglican | 1.114 | 0.568 | 0.770–1.611 |
| Pentecostal | 1.054 | 0.786 | 0.719–1.545 |
| **Living alone** | | | |
| Lives alone | 1.000 | | |
| Lives with others | 1.759 | *0.004* | 1.200–2.578 |
| **Rent status** | | | |
| Renting | 1.000 | | |
| Not renting | 1.096 | 0.549 | 0.811–1.482 |
| **Quality/durability of house** | | | |
| Well maintained | 1.000 | | |
| Fairly maintained | 0.754 | 0.080 | 0.550–1.034 |
| Dilapidated | 0.455 | *0.001* | 0.286–0.725 |
| **Household's main source of information** | | | |
| Radio | 1.000 | | |
| Word of mouth | 0.785 | 0.232 | 0.528–1.168 |
| Telephone | 1.432 | 0.086 | 0.950–2.160 |
| Television | 1.359 | 0.055 | 0.993–1.859 |
| **Type of water source** | | | |
| Improved source | 1.000 | | |
| Unimproved source | 0.584 | *0.044* | 0.346–0.985 |

indicated that older persons living with family members are at an advantage compared to those without since the elderly can count on the support in times of need [60].

There is variation in the source of water available to older persons living in the slum settlements. This ranges from improved sources (such as piped water) to unimproved ones (such as pond or wetland water). The lower likelihood of boiling water among older persons who were using unimproved sources is explicable, in part, in terms of social-economic environment in

which the older persons live [58]. It may be the case that the older persons who were using unimproved sources were also those who ranked low on the social economic ladder [61]. Such persons may have found difficulty affording means with which to treat the water before drinking it. Our finding also resonates with results from other related inquiries. For example, Daniel, Diener [11], found prevalence of linkages between household water treatment and socioeconomic status. Households which were connected to a piped water scheme had a higher probability of household water treatment compared to other types of water sources. The role of social economic status has similarly been found in previous studies where families with more household goods (refrigerator, stove, radio, and television) were significantly more likely to purify drinking water than those who had less goods [62]. The work of Mwabi, Adeyemo [63] on purification of water similarly dovetails with our findings.

We found that older persons who were residing in dilapidated houses were less likely to boil water for drinking compared to those who were staying in well maintained houses. This housing condition is an indicator of the low socio-economic status of older persons who were probably, limited in terms of resources, to help them afford sufficient energy for boiling of water, or, treating it using any other appropriate technology such as chlorination. This finding is in consonance with a study done in South Africa [64], which found that unsafely managed drinking water is associated with people's socio-economic status in slums. Similarly, a study done in Kenya [65] shows quality water for drinking is determined by an individual's socio-economic status. by Armah, Ekumah [45] indicates that urban rich persons have increased odds of drinking safely managed water than their counterparts who are poor. We recommend further research to be done to generate comprehensive and informative literature on housing condition and safe water drinking practices. This should be done to generate broader knowledge for appropriate policy interventions to be made with an aim of minimizing health risks (such as typhoid, diarrhea and others) associated with drinking unsafely managed water.

The study found that older persons who obtained water from unimproved sources were less likely to boil it to make it safe for drinking compared to those whose source was improved. This is linked to the perception that since they have been drinking water direct from the source there is no need to boil the it [66]. This perception can have detrimental effects on their lives and those of community members. Water sources get contaminated as a result of human and industrial activities that generate harmful waste into the water sources which makes an originally safe water source unsafe. It has been argued that a risky practice that has not previously resulted in illness is still a potential source of disease in the future [66]. According to a survey on Chinese healthy longevity, participants who drank unboiled water were more likely to develop cognitive impairment [67]. Another study conducted in China asserted that drinking unboiled or unsafe water by older persons was associated with mortality [68].

## Limitations

The study relied on self-reports of safe drinking water practices among older persons, which may be influenced by factors such as the desire to appear as though they were practicing recommended safe practices. This may lead to social desirability bias. Some older persons who may have not been boiling water for drinking could have reported that they were boiling it. There may also, be misreporting of information by older persons. This study was based on a question that required respondents to state whether they did anything to make the water safe for drinking. That is, we did not carry out bacteriological test to answer whether respondents really boil the water or not. Consequently, we did not delve deep into the dynamics of boiling

water which could have provided more insights. It is also possible that the existing household gender segregated roles could influence higher boiling water practice by older women than their male counterparts.

## Conclusions

Basing on our study findings, we find a disparity in boiling water practices to make it safe for drinking among older persons. For example, older women have increased odds of boiling water to make it safe for drinking compared to older men. Living arrangement is also a key factor of boiling water as living with others in households was beneficial for the older persons. The other predictors of not boiling water to make it safe for drinking among older persons were living in dilapidated house, unimproved water source and not getting financial support. Interventions targeting older men's involvement in safely managing drinking water should be underscored to minimize associated health risk and mortality resulting from drinking unsafe water in informal settlements. There is need for targeted information, education and communication on safe drinking water practices among older persons (especially through engaging with religious leaders). As government continues to help older persons especially by giving them financial assistance, it should also develop a culture of providing them with information and education on the benefits of boiling drinking water in their households. Further to the above, all stakeholders in Kampala including public health officials and political leaders are called upon to sensitize and mobilize older persons in their communities against drinking water that is potentially unsafe because no deliberate action is taken to make the water safe for dinking.

## Supporting information

**S1 Dataset.**
(DTA)

## Acknowledgments

We acknowledge Dr. Tobias C. Vogt from Faculty of Spatial Sciences, Department of Demography Landleven 1, University of Groningen, the Netherlands who reviewed the initial draft.

## Declarations

### Ethics approval and consent to participate

**Ethical approval:** Before data collection, efforts were made to meet ethical requirements for research in accordance with the research ethics guidelines [69]. Ethical clearance was obtained from the Ministry of Health Vector Control Division Research Ethic committee under number VCDREC 162 and Uganda National Council for Science and Technology (UNCST) HS2487ES. We obtained voluntary verbal informed consent from all the respondents before the commencement of each interview. Interviews were held in conditions and environments that ensured privacy of the respondents and the data were stored in a manner that did not allow access by an authorized persons. Also, the data storage removed any information such as names that can be used to identify the respondents.

## Author Contributions

**Conceptualization:** Fred Maniragaba, Abel Nzabona, Charles Lwanga, Paulino Ariho, Betty Kwagala.

**Data curation:** Fred Maniragaba, Paulino Ariho.

**Formal analysis:** Fred Maniragaba, Abel Nzabona, Charles Lwanga, Paulino Ariho.

**Funding acquisition:** Fred Maniragaba.

**Investigation:** Abel Nzabona, Paulino Ariho.

**Methodology:** Fred Maniragaba, Abel Nzabona, Paulino Ariho, Betty Kwagala.

**Project administration:** Fred Maniragaba.

**Resources:** Fred Maniragaba.

**Software:** Fred Maniragaba.

**Supervision:** Fred Maniragaba, Betty Kwagala.

**Validation:** Fred Maniragaba.

**Visualization:** Fred Maniragaba.

**Writing – original draft:** Fred Maniragaba, Abel Nzabona, Charles Lwanga, Paulino Ariho, Betty Kwagala.

**Writing – review & editing:** Fred Maniragaba, Charles Lwanga, Betty Kwagala.

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
