## [Decision Letter · Decision Letter 0]

30 May 2023

PONE-D-23-11738Safe water drinking practices among older persons in slums of Kampala: Analyzing gender disparities in boiling waterPLOS ONE

Dear Dr. Maniragaba,

Thank you for submitting your manuscript to PLOS ONE. After careful consideration, we feel that it has merit but does not fully meet PLOS ONE’s publication criteria as it currently stands. Therefore, we invite you to submit a revised version of the manuscript that addresses the points raised during the review process.

ACADEMIC EDITOR:  Dear Authors,

Thanks for your submission. I have some concerns:

1. I agree with the last general comments of the 3rd reviewer about gender, i.e., women is mainly responsible for water management so it raises a question whether your statistics results about older women are more likely to boil water than older man is valid or not. I think you need to state this as one of the main limitations of your study.

2. I also agree that your recommendations are not based on your findings.

3. In your revision, please include page and line number so it is easy for reviewers to locate their comments.

4. Please check your references. There is an error, e.g., no 12.

5. Introduction paragraph 6 -> you discuss boiling in the first 3 paragraphs, but then discuss older people in the rest of the paragraph, which is not related to the first 3 paragraphs.

6. The last paragraph of introduction talks about urbanizations/urban things, but you already focus on household water treatment (HWT) before. So the story line is poor. Please focus on the HWT itself rather than talk outside it.

7. I am confused now after reading your study objective, i.e., older person’s practices related to safe drinking water practices. But then you have also “gender disparities” in the title. They are two different things.

8. Also, safe drinking water practices are not only boiling/treatment, but also safe storage, which is not discussed in your study. So, I think you need to change the title.

9. It is still not clear for me why you only study older people. I can imagine that if there is younger woman/man at home, they will do the boiling and not the older people.

10. Your independent variables are general and not specifically related to gender, also from your results. So, maybe better if you remove “gender” from your title.

11. I think you can include policy implications in either discussion or conclusion section, and not in a separate section after conclusion. Also, I think better if you put the recommendation directly after you discuss the specific topic and not put them together in a special section. That will help reader to grasp whether your recommendations are in line with your findings and discussion or not.

Please consider carefully and incorporate all comments from reviewers in your revision.

We look forward to receiving your revised manuscript.

Kind regards,

D. Daniel, Ph.D.

Academic Editor

PLOS ONE

“**Funding: **This study was funded by the Consortium for Advanced Research Training in Africa (CARTA). CARTA is jointly led by the African Population and Health Research Center (APHRC) and the University of the Witwatersrand”

“Authors received no specific funding for this work”

Reviewers' comments:

Reviewer's Responses to Questions

**Comments to the Author**

1. Is the manuscript technically sound, and do the data support the conclusions?

Reviewer #1: Yes

Reviewer #2: Yes

Reviewer #3: No

2. Has the statistical analysis been performed appropriately and rigorously? 

Reviewer #1: Yes

Reviewer #2: Yes

Reviewer #3: Yes

3. Have the authors made all data underlying the findings in their manuscript fully available?

Reviewer #1: Yes

Reviewer #2: Yes

Reviewer #3: No

4. Is the manuscript presented in an intelligible fashion and written in standard English?

Reviewer #1: Yes

Reviewer #2: Yes

Reviewer #3: Yes

5. Review Comments to the Author

Reviewer #1: Title: Safe water drinking practices among older persons in slums of Kampala: Analyzing gender disparities in boiling water, Manuscript Number: PONE-D-23-11738

This is an interesting title and the way of writing is appreciated.

Still, there are comments and questions on this manuscript.

1. “The purpose of this study was to investigate the factors that influence boiling water practice …” so why not the topic include factors that influence…?

2. “The responses were coded as “yes” for those who reported that they boiled water…”, how do you categorize those respondents who boil water ‘rarely’, ‘sometimes’, ‘often’ etc. Yes/no, questions might not be appropriate to answer this research question.

3. This research question was good if supported by a bacteriological test to answer whether respondents really boil the water or not.

4. Your analysis was done using ‘multivariate’. Why do you prefer multivariate to multivariable some books/scholars stated that multivariate is for more than one outcome variable, but your response variable is one ‘boiling water (yes/no)’.

5. Why authors preferred the ‘Pearson Chi-squared test’ that binary logistic regression?

6. How does ‘quality of house’ act as a factor associated with ‘boil water’?

Reviewer #2: Overall, this is a well-written manuscript with a strong introduction with a well-informed study aim and study design. This manuscript only needs to undergo a few minor revisions.

The authors conduct very relevant research I recommend making sure to go back through to revise spelling and grammatical errors, such as extra spacing, punctuation, and instances of repeating words for example: in in as well as making sure the font is consistent in style and size throughout the text. I recommend reading through the manuscript again in entirety to catch where verb tense may be inconsistent or instances of incorrect noun-verb agreement as well.

I encourage the authors to provide more in-depth evidence and to cite additional studies in the discussion section to strengthen the arguments made, particularly in the second, fourth, and fifth paragraphs.

To improve readability, I would recommend restructuring the discussion section so the study implications and directions for future research are at the end together with more elaboration of why those recommendations would be effective based on previous research while citing more studies for the recommendations given.

Reviewer #3: General comments:

- The authors thoughtfully consider a vulnerable population that is, unfortunately, largely ignored in the WASH literature.

- Minor grammatical errors to be corrected

- Ensure consistency of significant figures throughout in text and tables

- One unfortunate limitation in using these data to make conclusions about water behaviors by gender is that it is well known that women are, by a wide margin, more likely to be the household water managers. Boiling household water is not necessarily an individual behavior – it only needs to be done by one person in the household to make the household’s drinking water safe, and that is typically a woman. Thus, the finding that women are more likely to boil water is to be expected, because they are more likely to have the chore of managing the household water that the men also drink. It does not appear that the data used in these analysis are specific enough that the authors are able to isolate individual behaviors around water treatment versus household division of labor along gendered lines. One could potentially ask, are female headed households more likely to boil their water versus male-headed households, or something along those lines, with these data. But, as analyzed currently, there is a blurring here between individual behaviors and "household-level" behaviors.

Specific comments:

Methods:

1. Because RR is a very common abbreviation for risk ratio or relative risk, I suggest avoiding using it as an unnecessary abbreviation for other terms.

2. I suggest removing the reference to the response rate formula – in this case it is a simple enough calculation.

3. Were households randomly selected from the identified households? Or were all households with older persons included? And were all older persons in each of these households eligible? Additionally, as described, it is not stratified (i.e., both divisions and parishes are randomly sampled and not first stratified by design). A stratified design implies sampling within defined strata.

4. Please clarify at the beginning of the methods that the data are from 593 and not 600 individuals, and that 600 individuals were invited to participate.

5. No comma is needed between October 2022.

6. In the “Variables” section:

a. The thorough description of the variables is appreciated. It is unclear how a “main” explanatory variable was selected – was this prespecified? Or were all of these variables considered equally? Please clarify.

b. It is unclear the difference between household “usual member” and “regular member”

c. Would it be correct to describe the rent status binary as “renting” or “owning”? That may be a clearer categorization.

d. Membership of Association could use additional description – I am not clear what this means.

e. Water source types should be separated by “or”

7. In the “Statistical analysis” section:

a. More detail is required to follow the weighting procedures and the under/over-sampling. Perhaps this indicates missing information in the prior section describing the sampling procedures.

b. Remove extraneous notes on statistics literature and tests – it will be clearer to just concisely describe your procedures/assumptions

c. The linktest function can help to check for specification errors, but, as far as I am aware, it is not a goodness of fit test. Please revise accordingly.

d. Overall, the modeling procedures are unclear. I understand that bivariate associations are being assessed with a chi-squared test. What is the variable selection procedure for the multivariate analysis?

8. Results

a. Ensure significant figures are consistent throughout (e.g., fix 65% in text versus 65.0% in the table)

b. From Table 1 -- I just want to confirm that 44 of the participants were children of the household head, and also, they were 60+ years of age?

c. From Table 1 -- since the majority of respondents do not live alone, could you please clarify what it means to get financial support (another explanatory variable)? Is this from the government?

d. From Table 2 and 3 – please revise p-value 0.000 to <0.001

e. Table 2 – it would be helpful to include sample sizes in the table. Please either add frequencies along with percentages, or consider combining Table 1 and 2.

f. If possible, include the % of otherwise eligible respondents who were excluded because they were sick (as noted in the Method exclusion criteria).

9. Discussion

a. The structure is clear, and the authors systematically discuss the findings.

b. Please revise the summary statement in the second sentence so that associated factors are correctly described. Suggest writing that boiling is “positively associated,” as that is accurate for most of the described factors. However, living in a dilapidated house and water from an unimproved source are negatively associated based on OR <1. (This is correctly discussed later on in the section.)

c. Revise “female older persons were nearly two times more likely” to “female older persons had twice the odds” – note the difference between a 2x likelihood (from a relative risk) versus 2x odds (from odds ratios)

d. A reference is needed for “…Catholic Church usually conducts health education…” sentence. The incorrect implication is that other religions do not.

e. In describing factors that are associated, it would improve the clarity to note whether what is being described is positively or negatively associated. For example, “Living with others was positively associated with boiling water…”

f. I appreciate that the authors point out the struggle to find time for boiling water – this is an important and underacknowledged barrier to household water treatment.

g. The discussion about living with others brings up questions about the specificity of the survey questions around boiling behavior. Can you attribute that behavior to the individual who is surveyed? Or does the question of boiling apply to the household’s behavior? And, if the latter, what if there are both males and females in the same household?

h. I am not clear what “real life analysis” means?

i. There are 2 separate paragraphs discussing the association with water source type – I suggest combining. Citations are needed in the discussion of perceptions of safety.

10. Policy implications

a. The authors note earlier that chlorination is another option for treating drinking water, and there are others. The results described here focus only on boiling, so the implications cannot be so broad as to imply that a lack of boiling is equivalent to unsafe water management. Make sure the conclusions are firmly grounded in the data and results.

6. PLOS authors have the option to publish the peer review history of their article (what does this mean?). If published, this will include your full peer review and any attached files.

Reviewer #1: No

Reviewer #2: **Yes: **Gabriela Stone

Reviewer #3: No

---

## [Author Response · Author response to Decision Letter 0]

12 Jul 2023

ACADEMIC EDITOR’s Comments: 

1. I agree with the last general comments of the 3rd reviewer about gender, i.e., women is mainly responsible for water management so it raises a question whether your statistics results about older women are more likely to boil water than older man is valid or not. I think you need to state this as one of the main limitations of your study.

Action: We appreciate this and we have taken the advice.

2. I also agree that your recommendations are not based on your findings.

Action: We revised this accordingly.

3. In your revision, please include page and line number so it is easy for reviewers to locate their comments.

Action: Thank you for your observation. Line numbers have been included in the revised manuscript.

4. Please check your references. There is an error, e.g., no 12.

Action: Thanks. We have revised this. 

5. Introduction paragraph 6 -> you discuss boiling in the first 3 paragraphs, but then discuss older people in the rest of the paragraph, which is not related to the first 3 paragraphs.

Action: We have addressed this in the revised manuscript.

6. The last paragraph of introduction talks about urbanizations/urban things, but you already focus on household water treatment (HWT) before. So the story line is poor. Please focus on the HWT itself rather than talk outside it.

Action: We have addressed this in the revised manuscript.

7. I am confused now after reading your study objective, i.e., older person’s practices related to safe drinking water practices. But then you have also “gender disparities” in the title. They are two different things.

Action: This was addressed in the revised manuscript as it was also raise by the reviewers.

8. Also, safe drinking water practices are not only boiling/treatment, but also safe storage, which is not discussed in your study. So, I think you need to change the title.

Action: We have revised the title to “Factors that influence safe water drinking practices among older persons in slums of Kampala: Analyzing disparities in boiling water”

9. It is still not clear for me why you only study older people. I can imagine that if there is younger woman/man at home, they will do the boiling and not the older people.

Action: We acknowledge this. We use living arrangement whether an older person lives alone with others as an independent variable.

10. Your independent variables are general and not specifically related to gender, also from your results. So, maybe better if you remove “gender” from your title.

Action: We have revised the title in the revised manuscript.

11. I think you can include policy implications in either discussion or conclusion section, and not in a separate section after conclusion. Also, I think better if you put the recommendation directly after you discuss the specific topic and not put them together in a special section. That will help reader to grasp whether your recommendations are in line with your findings and discussion or

 not.

Action: We have included the policy implications in in the conclusion section.

Reviewer #1: Title: Safe water drinking practices among older persons in slums of Kampala: Analyzing gender disparities in boiling water, Manuscript Number: PONE-D-23-11738

This is an interesting title and the way of writing is appreciated.

Still, there are comments and questions on this manuscript.

1. “The purpose of this study was to investigate the factors that influence boiling water practice …” so why not the topic include factors that influence…?

Action: Thank you. We have rephrased the title to reflect Factors that influence safe water drinking practices among older persons in slums of Kampala: Analyzing disparities in boiling water 

2. “The responses were coded as “yes” for those who reported that they boiled water…”, how do you categorize those respondents who boil water ‘rarely’, ‘sometimes’, ‘often’ etc. Yes/no, questions might not be appropriate to answer this research question.

Action: This is a good observation. However, the design of the question was binary, and we do not have any other options beyond ‘yes’ and ‘no’ in our question.

3. This research question was good if supported by a bacteriological test to answer whether respondents really boil the water or not.

Action: We agree with your observation and appreciate this nice insight. Because, ours purely based on self-reported responses, this has been added as one of the limitations of our study.

4. Your analysis was done using ‘multivariate’. Why do you prefer multivariate to multivariable some books/scholars stated that multivariate is for more than one outcome variable, but your response variable is one ‘boiling water (yes/no)’.

Action: The current narrative in the revised manuscript reflects ‘multivariable’ rather than ‘multivariate’.

5. Why authors preferred the ‘Pearson Chi-squared test’ than binary logistic regression?

Action: We appreciate that the unadjusted logistic regression can also be used as opposed to Pearson Chi square test. However, we used the Pearson Chi-squared test to isolate the factors which are significant at bivariate level since all the variables were categorical. In addition the variables which were significant at this level and those reported in literature were analyzed at the multivariable level using binary logistic regression. 

6. How does ‘quality of house’ act as a factor associated with ‘boil water’?

Action: The ‘quality of house’ was taken as a proxy indicator of socio-economic status of the respondents which may affect the capacity of the respondents to afford resources such as charcoal, electricity needed for boiling water.

Reviewer #2: Overall, this is a well-written manuscript with a strong introduction with a well-informed study aim and study design. This manuscript only needs to undergo a few minor revisions.

The authors conduct very relevant research I recommend making sure to go back through to revise spelling and grammatical errors, such as extra spacing, punctuation, and instances of repeating words for example: in in as well as making sure the font is consistent in style and size throughout the text. I recommend reading through the manuscript again in entirety to catch where verb tense may be inconsistent or instances of incorrect noun-verb agreement as well.

Action: Thanks. We have taken care of these concerns and also adhere to the journal guidelines. 

I encourage the authors to provide more in-depth evidence and to cite additional studies in the discussion section to strengthen the arguments made, particularly in the second, fourth, and fifth paragraphs.

Action: Thanks. We have taken care of these concerns

To improve readability, I would recommend restructuring the discussion section so the study implications and directions for future research are at the end together with more elaboration of why those recommendations would be effective based on previous research while citing more studies for the recommendations given.

Action: Thank you for your comment. We have endeavored to follow Plos One format. 

Reviewer #3: General comments:

- The authors thoughtfully consider a vulnerable population that is, unfortunately, largely ignored in the WASH literature.

- Minor grammatical errors to be corrected

- Ensure consistency of significant figures throughout in text and tables

Action: We appreciate your observation. We have taken care of this.

- One unfortunate limitation in using these data to make conclusions about water behaviors by gender is that it is well known that women are, by a wide margin, more likely to be the household water managers. Boiling household water is not necessarily an individual behavior – it only needs to be done by one person in the household to make the household’s drinking water safe, and that is typically a woman. Thus, the finding that women are more likely to boil water is to be expected, because they are more likely to have the chore of managing the household water that the men also drink. It does not appear that the data used in these analysis are specific enough that the authors are able to isolate individual behaviors around water treatment versus household division of labor along gendered lines. One could potentially ask, are female headed households more likely to boil their water versus male-headed households, or something along those lines, with these data. But, as analyzed currently, there is a blurring here between individual behaviors and "household-level" behaviors.

Action: We agree with this line of thought. Indeed, we don’t have sufficient data to distinguish between individual level behaviors and household level behaviors. Consequently we have adjusted and reported more appropriately in the revised manuscript.

Specific comments:

Methods:

1. Because RR is a very common abbreviation for risk ratio or relative risk, I suggest avoiding using it as an unnecessary abbreviation for other terms.

Action: Thanks for your observation. We have stopped abbreviating it.

2. I suggest removing the reference to the response rate formula – in this case it is a simple enough calculation.

Action: We have removed the narrative and reference to the response rate formula.

3. Were households randomly selected from the identified households? Or were all households with older persons included? And were all older persons in each of these households eligible? Additionally, as described, it is not stratified (i.e., both divisions and parishes are randomly sampled and not first stratified by design). A stratified design implies sampling within defined strata.

i. Were households randomly selected from the identified households? 

Action: Yes, the households were randomly selected. 

ii. Were all households with older persons included? 

 Action: No. the threshold was informed by the determined sample size. In addition, because of the simple random sampling mechanism we used, some households were left out.

iii. Were all older persons in each of these households eligible? 

Action: The only selection criteria was age 60, so, any person who met this criteria was eligible.

iv. Additionally, as described, it is not stratified (i.e., both divisions and parishes are randomly sampled and not first stratified by design). A stratified design implies sampling within defined strata.

Action: We corrected this in the revised manuscript.

4. Please clarify at the beginning of the methods that the data are from 593 and not 600 individuals, and that 600 individuals were invited to participate.

Action: Thanks for this comment. We have clarified the statement at the beginning of the paragraph. 

5. No comma is needed between October 2022.

Action: We have edited and removed the comma. 

6. In the “Variables” section:

a. The thorough description of the variables is appreciated. It is unclear how a “main” explanatory variable was selected – was this prespecified? Or were all of these variables considered equally? Please clarify.

Action: We have rephrased the statement to indicate that the variables had equal consideration.

b. It is unclear the difference between household “usual member” and “regular member”

Action: A usual member is a person who has been living in the household for the last six month or more during the last 12 month before our survey. A ‘regular’ member is a person who would have been a usual member but has only stayed for less than six month in the household at the time of our survey.

c. Would it be correct to describe the rent status binary as “renting” or “owning”? That may be a clearer categorization.

Action: We appreciate this comment. We however note that owning does not necessarily mean the opposite of renting because some people may be overseers or free users of the dwelling units. We have therefore categorized the variable as ‘renting’ or not ‘renting’.

d. Membership of Association could use additional description – I am not clear what this means.

Action: Thanks for this comment. Membership of association means belonging to associations such as cooperative saving societies, church associations and others.

e. Water source types should be separated by “or”

Action: We have corrected this write up.

7. In the “Statistical analysis” section:

a. More detail is required to follow the weighting procedures and the under/over-sampling. Perhaps this indicates missing information in the prior section describing the sampling procedures.

Action: We thank you for this comment. We have addressed this in the revised manuscript.

b. Remove extraneous notes on statistics literature and tests – it will be clearer to just concisely describe your procedures/assumptions

Action: We have addressed this as advised.

c. The linktest function can help to check for specification errors, but, as far as I am aware, it is not a goodness of fit test. Please revise accordingly.

Action: We thank you for this observation. We have revised accordingly.

d. Overall, the modeling procedures are unclear. I understand that bivariate associations are being assessed with a chi-squared test. What is the variable selection procedure for the multivariate analysis?

Action: We thank you for this comment. Variables that were significant at bivariate level were analyzed at the multivariate level. Nonetheless, those that were not significant at bivariate such as age, but deemed to be relevant to Socio-economic status of older persons as informed by prior studies were included in the model.

8. Results

a. Ensure significant figures are consistent throughout (e.g., fix 65% in text versus 65.0% in the table)

Action: We have revised by adding decimals in the text

b. From Table 1 -- I just want to confirm that 44 of the participants were children of the household head, and also, they were 60+ years of age?

Action: We confirm that the 44 participants were aged 60+ years of age. The head of the household was older 60+ years. For instance, some of the female older persons were staying with their older parents as dependents.

c. From Table 1 -- since the majority of respondents do not live alone, could you please clarify what it means to get financial support (another explanatory variable)? Is this from the government?

Action: We appreciate this observation. The majority of the older persons did not live alone as some stayed with children, others with young grandchildren and some with hired house helpers. The financial support came from many sources including government (social assistance grant for empowerment), children and others.

d. From Table 2 and 3 – please revise p-value 0.000 to <0.001

Action: We have revised this.

e. Table 2 – it would be helpful to include sample sizes in the table. Please either add frequencies along with percentages, or consider combining Table 1 and 2.

Action: We have added the sample size in Table 2 of the revised manuscript.

f. If possible, include the % of otherwise eligible respondents who were excluded because they were sick (as noted in the Method exclusion criteria).

Action: We appreciate this. However, we did not count those who were sick because we proceeded to the next potential respondent.

9. Discussion

a. The structure is clear, and the authors systematically discuss the findings.

Action: We appreciate this compliment

b. Please revise the summary statement in the second sentence so that associated factors are correctly described. Suggest writing that boiling is “positively associated,” as that is accurate for most of the described factors. However, living in a dilapidated house and water from an unimproved source are negatively associated based on OR <1. (This is correctly discussed later on in the section.)

Action: We appreciate this comment. We have revised accordingly.

c. Revise “female older persons were nearly two times more likely” to “female older persons had twice the odds” – note the difference between a 2x likelihood (from a relative risk) versus 2x odds (from odds ratios)

Action: We have revised this according. We appreciate the advice.

d. A reference is needed for “…Catholic Church usually conducts health education…” sentence. The incorrect implication is that other religions do not.

Action: We thank the reviewer for this observation. We have addressed this in the revised manuscript.

e. In describing factors that are associated, it would improve the clarity to note whether what is being described is positively or negatively associated. For example, “Living with others was positively associated with boiling water…”

Action: We appreciate this observation. We have revised this

f. I appreciate that the authors point out the struggle to find time for boiling water – this is an important and underacknowledged barrier to household water treatment.

Action: We thank the reviewer for the complement

g. The discussion about living with others brings up questions about the specificity of the survey questions around boiling behavior. Can you attribute that behavior to the individual who is surveyed? Or does the question of boiling apply to the household’s behavior? And, if the latter, what if there are both males and females in the same household?

Action: During the survey, the respondent was asked “Do you do anything to make your drinking water safe?” Boiling was the most common method of making water safe for drinking. Therefore, our question sought to address individual behavior on the boiling of drinking water.

h. I am not clear what “real life analysis” means?

Action: We have revised this statement so that it is clearer

i. There are 2 separate paragraphs discussing the association with water source type – I suggest combining. Citations are needed in the discussion of perceptions of safety.

Action: We have combined the paragraphs. We have added a citation. 

10. Policy implications

a. The authors note earlier that chlorination is another option for treating drinking water, and there are others. The results described here focus only on boiling, so the implications cannot be so broad as to imply that a lack of boiling is equivalent to unsafe water management. Make sure the conclusions are firmly grounded in the data and results.

Action: We have revised this statement and focused only on the boiling water as a cheap and efficient method of making water safe for drinking especially in the informal settlements.

---

## [Decision Letter · Decision Letter 1]

10 Aug 2023

PONE-D-23-11738R1Factors that influence safe water drinking practices among older persons in slums of Kampala: Analyzing disparities in boiling waterPLOS ONE

Dear Dr. Maniragaba,

Thank you for submitting your manuscript to PLOS ONE. After careful consideration, we feel that it has merit but does not fully meet PLOS ONE’s publication criteria as it currently stands. Therefore, we invite you to submit a revised version of the manuscript that addresses the points raised during the review process.

We look forward to receiving your revised manuscript.

Kind regards,

D. Daniel, Ph.D.

Academic Editor

PLOS ONE

Journal Requirements:

Additional Editor Comments:

Dear Author,

Thank you for your revision. There are still some comments from one reviewer, please adjust your draft accordingly.

Reviewers' comments:

Reviewer's Responses to Questions

**Comments to the Author**

1. If the authors have adequately addressed your comments raised in a previous round of review and you feel that this manuscript is now acceptable for publication, you may indicate that here to bypass the “Comments to the Author” section, enter your conflict of interest statement in the “Confidential to Editor” section, and submit your "Accept" recommendation.

Reviewer #2: All comments have been addressed

2. Is the manuscript technically sound, and do the data support the conclusions?

Reviewer #2: Yes

3. Has the statistical analysis been performed appropriately and rigorously? 

Reviewer #2: Yes

4. Have the authors made all data underlying the findings in their manuscript fully available?

Reviewer #2: Yes

5. Is the manuscript presented in an intelligible fashion and written in standard English?

Reviewer #2: Yes

6. Review Comments to the Author

Reviewer #2: Line 244-246 there is a statement from this article link that needs to be paraphrased as it is word for word from the original source at https://www.ncbi.nlm.nih.gov/pmc/articles/PMC7115273/ : "Relatedly, Allegranzi, Memish (50), argue that religious faith has made many important contributions to the ethics of health care and has helped focus the attention of health care providers on both the physical and spiritual nature of humans."

Minor grammatical and spelling errors remain present throughout the manuscript.

Line 363-365 discusses the predictors of boiling water, however the wording is unclear. I would recommend rephrasing it for consistency of your findings. "The other predictors of not boiling water to make it safe for drinking..." since older individuals living in a dilapidated house, unimproved water sources, and not receiving financial support were less likely to boil their drinking water.

Overall, the authors took time to implement the revisions suggested in the previous round which made this paper much stronger. All of my other suggestions and those by the previous reviewers have been adequately addressed.

7. PLOS authors have the option to publish the peer review history of their article (what does this mean?). If published, this will include your full peer review and any attached files.

Reviewer #2: **Yes: **Gabriela Stone

---

## [Author Response · Author response to Decision Letter 1]

1 Sep 2023

Reviewer #2: Line 244-246 there is a statement from this article link that needs to be paraphrased as it is word for word from the original source at https://www.ncbi.nlm.nih.gov/pmc/articles/PMC7115273/ : "Relatedly, Allegranzi, Memish (50), argue that religious faith has made many important contributions to the ethics of health care and has helped focus the attention of health care providers on both the physical and spiritual nature of humans."

Action: We thank the reviewer for this important observation. We have paraphrased this.

Minor grammatical and spelling errors remain present throughout the manuscript.

Action: We appreciate this comment. We have read through the manuscript and corrected accordingly.

Line 363-365 discusses the predictors of boiling water, however the wording is unclear. I would recommend rephrasing it for consistency of your findings. "The other predictors of not boiling water to make it safe for drinking..." since older individuals living in a dilapidated house, unimproved water sources, and not receiving financial support were less likely to boil their drinking water.

Action. Thanks for this suggestion. This has been addressed as suggested

Overall, the authors took time to implement the revisions suggested in the previous round which made this paper much stronger. All of my other suggestions and those by the previous reviewers have been adequately addressed.

Action: We appreciate this.

---

## [Editor Report · Decision Letter 2]

10 Sep 2023

Factors that influence safe water drinking practices among older persons in slums of Kampala: Analyzing disparities in boiling water

PONE-D-23-11738R2

Dear Dr. Maniragaba,

We’re pleased to inform you that your manuscript has been judged scientifically suitable for publication and will be formally accepted for publication once it meets all outstanding technical requirements.

Kind regards,

D. Daniel, Ph.D.

Academic Editor

PLOS ONE

Additional Editor Comments (optional):

Thank you for addressing all reviewer's comments appropriately
---

## [Editor Report · Acceptance letter]

13 Sep 2023

PONE-D-23-11738R2 

Factors that influence safe water drinking practices among older persons in slums of Kampala: Analyzing disparities in boiling water 

Dear Dr. Maniragaba:

I'm pleased to inform you that your manuscript has been deemed suitable for publication in PLOS ONE. Congratulations! Your manuscript is now with our production department. 

Kind regards, 

on behalf of

Mr D. Daniel 

Academic Editor

PLOS ONE